# The Effects of Plasma-Activated Water on Heavy Metals Accumulation in Water Spinach

Chih-Yao Hou [1] , Ting-Khai Kong [2], Chia-Min Lin [1,†] and Hsiu-Ling Chen [2,*,†]

1 Department of Seafood Science, National Kaohsiung University of Science and Technology, Kaohsiung 811, Taiwan; chihyaohou@nkust.edu.tw (C.-Y.H.); cmlin@mail.naku.edu.tw (C.-M.L.)
2 Department of Food Safety/Hygiene and Risk Management, College of Medicine, National Cheng Kung University, Tainan 701, Taiwan; 10912015@gs.ncku.edu.tw
* Correspondence: hsiulinchen@mail.ncku.edu.tw
† The authors has the same contribution.

**Abstract:** Toxic heavy metals accumulate in crops from the environment through different routes and may interfere with biochemical reactions in humans, causing serious health consequences. Plasma technology has been assessed for the promotion of seed germination and plant growth in several past studies. Therefore, the aim of the present study was to evaluate whether the growth rate of plants can be increased with the application of non-thermal plasma, as well as to reduce the accumulation of heavy metals in leafy vegetables (water spinach). In this study, several kinds of plasma treatments were applied, such as treatment on the seeds (PTS + NTW), irrigation water (NTS + PAW) or both (PTS + PAW). The results of the study showed that the heavy metals accumulated in water spinach were affected by the heavy metals available in the soil. The bioconcentration factor (BCF) of Cd in water spinach decreased from 0.864 to 0.543 after plasma treatment in seed or irrigating water, while the BCF of Pb was low and did not show any significant changes. Therefore, the results suggest that plasma treatment may suppress Cd absorption, but not for Pb. In this study, plasma treatment did not help to improve the product yield of water spinach planted in Cd-added soil. In the future, fertilizers can be used to supply nutrients that are not provided by plasma-activated water to support the growth of water spinach.

**Keywords:** plasma-activated water; heavy metals; water spinach; plant growth





## 1. Introduction

Advancements in technology, industrialization, and urbanization have sped up the pollution of the environment. Potential toxic heavy metals, including lead (Pb), cadmium (Cd), mercury (Hg), arsenic (As), nickel (Ni), chromium (Cr), copper (Cu), zinc (Zn), cobalt (Co), and other metal elements in the environment are emitted into water and soil through industrial wastewater discharge, commercial waste disposal and burial, exhaust emissions, and the improper use of agricultural materials. Toxic heavy metals are less mobile in the environment, and they possess a high residual capacity, and hence are easily accumulated in the environment. Heavy metal uptake can occur through mechanisms such as phytoextraction, phytostabilization, rhizofiltration, and phytovolatilization, which is the uptake, absorption, translocation, and/or transpiration process of metals by the plant into the above-ground portion of the roots [1]. Most of the heavy metals in soil can accumulate in crops, and consumption of crops that are contaminated with heavy metals can pose serious human health issues and present some adverse effects [2]. In Taiwan, the content of the heavy metals Cd and Pb in leafy vegetables should not exceed 0.2 mg/kg F.W. and 0.3 mg/kg F.W., respectively, according to the Taiwan Food and Drug Administration's regulations.

Plasma is considered to be a state of matter other than solid, liquid, and gas. Basically, plasma can be generated by subjecting different types of gases to an electric or electromagnetic field, which provides sufficient energy for ionization, or dissociation or exciting collisions. The increased number of collisions between the electrons due to the magnetic field causes an increase in the energy in the ionized gas. As a result, a quasi-neutral gas is formed that contains different species, such as electrons, ions, radicals, atoms, and molecules in their fundamental or excited states, with a net neutral charge [3]. Plasma can be categorized into high-temperature plasma (equilibrium plasma), thermal plasma (quasi-equilibrium plasma), and non-thermal plasma (non-equilibrium plasma) [4]. If the plasma ions and neutral components remain at or near room temperature, it is considered to be low-temperature or non-equilibrium plasma. The special characteristics of plasma with a strong thermodynamic non-equilibrium nature and low gas temperature include containing a lot of reactive chemical species, which makes non-thermal plasma applicable to several different industries [5]. Therefore, the application of plasma in agriculture is one of the most promising applications being studied in recent years. The main research direction in agricultural applications and food safety are as follows: inducing changes in the plant characteristics, eliminating pathogens or inhibiting microorganisms, promoting seed germination, increasing crop quantity, and improving crop preservation, and pesticide elimination [6–9].

Pawłat et al. [10] applied an atmospheric pressure plasma jet that operated with dielectric barrier discharge (DBD) on *Lavatera thuringiaca* L. seeds' germination at different exposure times (1, 2, 5, 10, and 15 min). The experimental results revealed that the germination capacity and germination energy of the seeds in the experimental group were improved as a result of plasma stimulation before sowing. However, excessive exposure can cause damage to the seeds, resulting in decreased germination. The germination rate, germination vigor, relative conductivity, and water uptake of the oilseed rape were improved, while the plant yield was increased by 28.2% after the seeds were treated by a low-vacuum helium cold plasma at a parameter of 100 W, 13.56 MHz, and 150 Pa for 15 s [11]. Research on direct and indirect plasma treatments of several vegetable species with DBD was conducted by Liu et al., who found that the germination of seeds was improved, although the efficiency depended on the species [12].

Ma et al. proved that the polysaccharides of a mutant strain of *Ganoderma lingzhi mycelia* increased by 25.6% after treatment with atmospheric DBD non-thermal plasma [7]. Another similar study by Takaki et al. included the application of a pulsed high-voltage electrical stimulation to cultivation mushrooms at 50–130 kV, where the fruiting body formation increased by 1.3–2 times [9].

Plasma-treated water or plasma-activated water (PAW) is produced by applying plasma to water in order to generate active oxygen molecules and excited OH radicals. Atomic oxygen and nitrogen generated in gaseous plasma could transfer into liquid and be easily converted into other reactive oxygen species (ROS) and reactive nitrogen species (RNS), such as nitrates ($NO_3^-$), nitrites ($NO_2^-$), and hydrogen peroxides ($H_2O_2$) in the solution. Air is considered to be the most abundant processing gas at low cost, and acts as a great source of oxygen ($O_2$) and nitrogen ($N_2$) with ultraviolet radiation and other chemical species in liquids. PAW is a mixture of high biochemically reactive solution of an acidic condition due to the changes in oxidation-reduction potential (ORP), electrical conductivity (EC), and the formation of RON and RNS [13,14]. Judée et al. evaluated the potential application of plasma treatment of tap-water (PATW) produced by cold atmospheric pressure DBD for improving agricultural quality [15]. The study revealed that the formation of nitrite, nitrate, ammonium ions ($NH_4^+$), and $H_2O_2$ in the water could induce and enhance crop growth, and that the consumption of bicarbonate ions could promote seedling growth. The length of the seedlings increased by 34% and 128.4% after 3 and 6 days of PATW treatment. The production of reactive species using plasma-activated water could change or affect the seeds both physically and chemically. Thirumdas et al. reviewed the relevant research on PAW and indicated that the physical and chemical

properties of PAW are related to the acidity, conductivity, redox potential, and concentration of ROS and RNS in the water, which affects microorganisms [14,16]. In addition, Thirumdas et al. suggested that PAW has a synergistic effect on food disinfection and can also promote the growth of seed seedlings [14]. The increment in the proportion of nitrate and nitrite ions in PAW may be the actual reason for the improvement in plant growth. Soaking seeds in PAW can play an antibacterial role and can also promote seed germination and plant growth. PAW may be used to increase crop yields and confront drought environmental conditions [12,14].

Several previous studies in the literature have proven the efficiency of plasma treatment for improving the germination and growth rate of plant species. Hence, this research focuses on assessing the application of non-thermal plasma in decreasing the heavy metal accumulation of vegetables planted in contaminated soil by initiating the growth of the vegetables.

## 2. Materials and Methods

### 2.1. Chemicals

Cadmium (II) acetate anhydrous (Sigma-Aldrich, India, 99.995%), lead acetate trihydrate (Sigma-Aldrich, India, 99–102.5%), ICP multi-element standard solution VI (Merck, Germany, 1000 mg/L), rhodium ICP standard (Merck, Germany, 1000 mg/L), nitric acid (JT Baker, Canada, 70%), nitric acid 65% for analysis (Merck, Germany), and hydrochloric acid fuming 37% for analysis (Merck, Germany) were used in the current study.

### 2.2. Man-Made Contaminated Soil

For preparing Cd-contaminated soil and Pb-contaminated soil, we dissolved 12.81 g of lead (II) acetate in 1 L of water and mixed them into 10 kg of cultivation soil, purchased from the gardening market (SINON CORPORATION, Tainan, Taiwan) to prepare 700 mg/kg wet weight (W.W.) man-made contaminated soil. For Cd-contaminated soil, 0.14 g of cadmium acetate was weighed and dissolved in 1 L of water, followed by mixing with 10 kg of cultivated soil. The concentration was then expected to be 7 mg/kg W.W. The man-made contaminated soil was continuously mixed well for 1 week to ensure that the metals were stable and uniformly distributed.

### 2.3. Selection of Vegetables

Water spinach (*Ipomoea aquatica*) seeds were selected and purchased (Known-You Seed Co., LTD, Kaohsiung, Taiwan), as leafy vegetables are relatively more frequently consumed by the Taiwanese population, on the basis of data released by the National Food Consumption Database.

### 2.4. The Plasma Device and Parameters

The non-thermal atmospheric plasma generator was designed and produced by the Aerothermal and Plasma Physics Lab (APPL), Department of Mechanical Engineering, National Chiao Tung University (Hsinchu, Taiwan). This system included a high-voltage power supply, an air pump, and an atmospheric pressure plasma jet (APPJ) (patent US10,121,638B1) generator [17]. This system was specifically designed to generate plasma beneath the water surface. Air was pumped into the electrode by the air pump under a regular atmosphere at a flow rate of 10 standard liters per minute (slm). The input air was ionized by the high-voltage electrode to generate plasma, which was then delivered through a quartz capillary into the water. Reverse osmotic (RO) water was used as the water source. The input power and frequency for the power supply were AV 110 V and 60 Hz, respectively. The output voltage, frequency, and power were set at 3.0 kV, 16 kHz, and 60 Watts, respectively.

### 2.5. Plasma Treatment of Seeds and Irrigation Water

The first step in the plasma application was to prepare PAW. A total of 500 mL of reverse osmosis (RO) water was prepared in a 1000-mL beaker. Two plasma jets were immersed in water, and the beaker was covered with a plastic wrap. The power was set to 60 W for plasma generation and the water was treated for 20 min. For the control group, RO water without plasma treatment was used as the irrigation water for plantation purposes. For seed treatment, 15 seeds were selected randomly and placed in a beaker containing 200 mL of RO water. One plasma jet was immersed in water, and the power was set to 60 W for plasma generation. The seeds were treated for 7 min after a preliminary study revealed that this was optimal for achieving high efficiency in germination. For the control group, the seeds were placed in the water for 7 min without any plasma application.

### 2.6. Plantation of Water Spinach

Planting pots (15 cm diameter, 13 cm height) were used for plantation purposes. Clean cultivation soil (control group) and man-made contamination soil with added metals were weighed. A total of 550 g of soil was placed into each plantation pot. For seedling preparation, the treated and untreated seeds were then placed on wet filter paper in a Petri dish for a germination period of 5 days. Six of the best-grown seedlings were selected and moved to each pot for growing. The plasma treatment was applied to the seeds and the water, in combinations as follows: Group 1, no plasma treatment on seeds or irrigation water (NTS + NTW); Group 2, no plasma treatment on seeds, but with the use of PAW as irrigation water (NTS + PAW); Group 3, plasma treatment on seeds, but no plasma treatment was used on irrigation water (PTS + NTW); and Group 4, plasma treatment on seeds, and using PAW as the irrigation water (PTS + PAW). Each group was planted in three pots (triplicate), and each pot contained six seedlings. The procedure was shown in Figure 1. The plantation was carried out for 5 weeks under neutral day conditions (photoperiod: 13-h light between 7:00 a.m. to 8:00 p.m.) with the aid of a T9 fluorescent lamp (6500 K, 18 W). Each pot of water spinach was watered with 150 mL PAW or RO water thrice each week, while the plant height measurement and photoshoot were performed on the 4th day of each week.

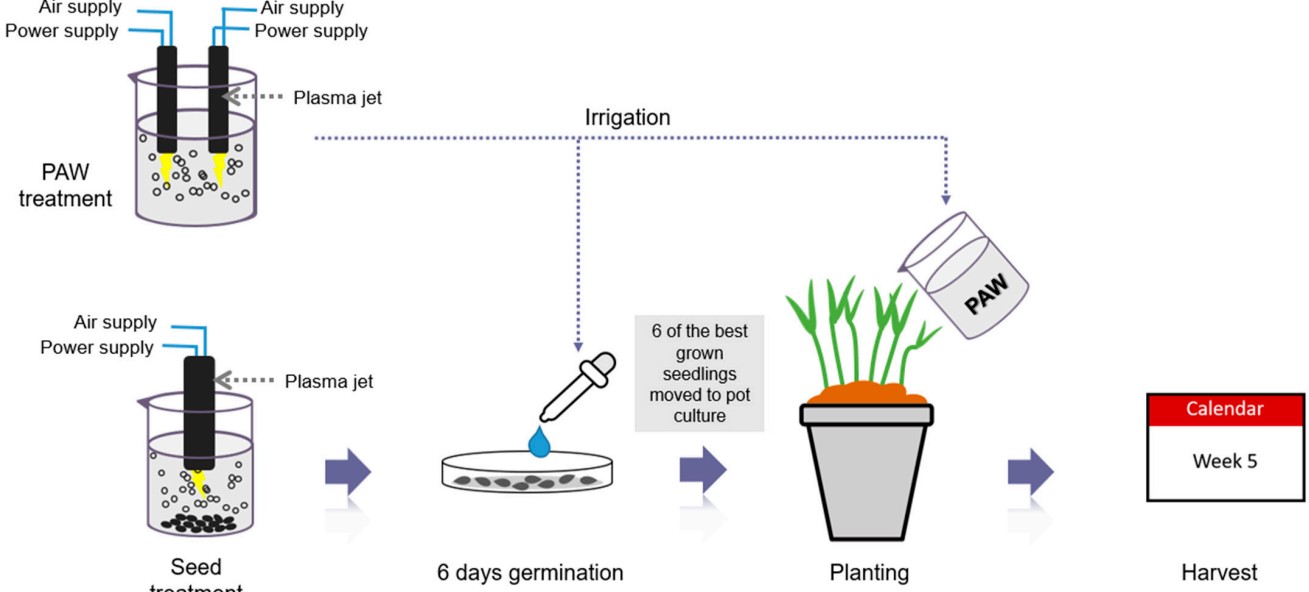

**Figure 1.** The procedure of the plantation.

*2.7. Physiochemical Properties Analysis*

2.7.1. Determination of Nitrates and Nitrites of PAW

The $NO_3^-$ and $NO_2^-$ contents of water samples were determined by NitraVer® 4 Nitrate Reagent Powder Pillow and NitraVer® 2 Nitrite Reagent Powder Pillow, respectively, with the aid of DR900 Multiparameter Portable Colorimeter, according to the instructions provided by the manufacturer (HACH, Loveland, CO, USA).

2.7.2. Determination of pH, Oxidation-Reduction Potential (ORP) and Electrical Conductivity (EC), and $H_2O_2$ of PAW

Physicochemical properties such as pH, ORP, and EC were determined immediately after the generation of PAW. The pH values and conductivity of the PAW were evaluated using a pH electrode and electro-conductivity probe (Cole-Parmer PC200-2 pH/Conductivity Meter Kit). On the other hand, the redox potential was measured using an SP-2100 Laboratory pH/ORP meter with the corresponding ORP probe. $H_2O_2$ was measured using a Hydrogen Peroxide Test Kit (Model HYP-1, product no.2291700) (HACH, USA).

*2.8. Metals Analysis*

2.8.1. Preparation of Soil Samples

The cultivated soil samples were collected at the beginning and the end of the plantation from each pot. A PET bottle was used to store the samples. Before the freeze-drying process, the samples were stored at $-18$ °C in a freezer for 24 h. The freeze-drying process of the vegetable samples and soil samples lasted for between 48 and 72 h. For the soil samples, a mortar and pestle were used to grind the samples into a fine powder, and the samples were then filtered through a sifting screen.

2.8.2. Determination of Heavy Metals in Soils

This sample pretreatment refers to Microwave-Assisted Aqua Regia Digestion Method for the Analysis of Heavy Metals in Soil (NIEA S301.61B). First, 0.25 g of the sample was placed into microwave digestion vessels. Three milliliters of 37% HCl and 1 mL 65% $HNO_3$ were then added to the vessels, and the mixture was allowed to react for 30 min. The mixtures were then digested with a microwave oven (MarsXpress microwave system, CEM) at 1800 W. The digested mixtures were then filtered with the Hydrophilic PTFE Filter (30 mm $\times$ 0.45 μm) and additional deionized water was added to make the volume of the mixtures 50 mL. The determination was performed in triplicate. The samples, blank samples, and standard solutions with different concentrations were analyzed with Inductively Coupled Plasma Optical Emission Spectrometer (ICP-OEs, PerkinElmer Avio 200, San Jose, CA, USA). Detected concentrations in samples that were lower than the method detection limit (MDL) are expressed as ND, and <LOQ is stated for detected concentrations that were lower than the limit of quantification.

2.8.3. Preparation of Vegetable Samples

The edible portions of the water spinach were cut at a position around 1 cm above the soil surface for chemicals analysis. The harvested samples were weighed and then cleaned using RO water to remove the dirt. The water spinach samples were dried with paper towels and placed in zipper bags with the corresponding labels. Samples collected from the same treatment combination were maintained in the same zipper bag to make it a mixed sample from three pots of a plant. Before the freeze-drying process, the samples were stored at $-18$ °C in the freezer for 24 h. The freeze-drying process of the vegetable samples and soil samples lasted for between 48 and 72 h. A homogenized blender was used to chop the vegetable samples into smaller pieces.

2.8.4. Determination of Heavy Metals in Vegetables

First, 0.3 g of the samples were weighed and placed into microwave digestion vessels. Then, 3 mL of 67% $HNO_3$ was added to the vessels and mixed with the samples.

The mixtures were then digested with a microwave oven (MarsXpress microwave system, CEM) at 1800 W. The digested mixtures were then filtered with the Hydrophilic PTFE Filter (30 mm × 0.45 μm) and additional deionized water was added to make the volume of the mixtures 25 mL for further analysis. The samples, blank samples, and standard solutions with different concentrations were then analyzed with Inductively Coupled Plasma Mass Spectrometry (ICP-MS; PerkinElmer NexION 2000, San Jose, CA, USA).

### 2.8.5. Quality Assurance/Quality Control for Metal Analysis

The recovery efficiency tests for Pb and Cd were conducted using the same sample analysis procedure with the addition of a standard solution. The recovery rates ranged from 103% and 89% for Pb and Cd in the soil samples and 108% and 93% for Pb and Cd in vegetables in the present study. The method detection limit (MDL) was used with a concentration that was slightly lower than the lowest concentration of the calibration curve. Measurements at this concentration were repeated seven times to estimate the standard deviation, and MDL was set as three times the standard deviation. The MDL ranged from 0.047 μg/g and 0.082 μg/g for Pb and Cd g in soil and 0.230 μg/g and 0.132 μg/g for Pb and Cd in vegetables.

### 2.9. Bioconcentration Factor

Bioconcentration factor (BCF) is described as the ability of plants to absorb heavy metals from contaminated soils. The BCF was calculated as suggested by previous studies [18–20]:

$$\text{BCF} = \frac{\text{Heavy metal concentration in vegetables}}{\text{Heavy metal concentration in soil}} \tag{1}$$

### 2.10. Statistical Analysis

The data were analyzed by the IBM SPSS Statistics 23. All determinations in this study were performed in triplicate. The significant differences in the concentration of heavy metals in soil were analyzed by Kruskal–Wallis test. The results were presented as mean ± standard deviation. $p = 0.05$ indicated statistical significance.

## 3. Results and Discussion

### 3.1. Physiochemical Properties of Irrigation Water

The generation of PAW can be confirmed by analyzing the acidity, conductivity, ORP, and concentration of ROS and RNS in the water [14]. Thirumdas et al. mentioned that the increase in the $NO_3^-$ and $NO_2^-$ contents in PAW could be the main reason behind enhanced plant growth and seed germination [14]. In this study, RO water was used as the water source and treated for 5, 10, 15, and 20 min to generate PAW for the selection of the most favorable PAW as irrigation water. The physiochemical characteristics of PAW under different treatment times are summarized in Table S1. In a nutshell, the $NO_3^-$ and $NO_2^-$ concentrations, as well as the EC, ORP and $H_2O_2$ values, increased with increasing treatment time, except that the pH values of PAW decreased with increasing treatment time (Figures S1–S4).

Figure 2 illustrates the concentration of $NO_3^-$ generated by plasma in this study. The concentration of $NO_3^-$ increased from untreated (control) to 20-min treatment. The concentration of $NO_3^-$ decreased in the order 20 min > 15 min > 10 min > 5 min > control. The increasing trend nitrate concentration with increasing treatment time was similar to results reported in some other studies [21,22]. The findings of Zhou et al. [21] indicated that $NO_2^-$ and $NO_3^-$ are formed in water, with the constant rate following zero-order kinetics, indicating a direct effect of the plasma. Meanwhile, they also reported that air micro-plasma treatment resulted in the highest concentration of nitrites and nitrates, followed by nitrogen plasma and oxygen plasma. Li et al. [22] also mentioned that the concentrations of $NO_2^-$, $NO_3^-$, and $H_2O_2$ increased with increasing discharge current or activation time [22]. Figure 3 depicts the concentrations of $NO_2^-$ in PAW treated for different lengths

of time in this study. However, the $NO_2^-$ concentration of PAW in this study do not follow the time-dependent trend with treatment suggested by other researchers. $NO_2^-$ was not detected in RO water, but it was present when water dissolved the ions resulting from the breakdown of nitrogen and oxygen in the plasma treatment. The concentration of $NO_2^-$ was highest after 10-min PAW treatment (PAW10), followed by PAW20 > PAW15 > PAW5. Furthermore, some studies in the literature have reported that no $NO_2^-$ was detected after 1 and 5 min of plasma activation [23]. $NO_3^-$ plays important roles as the nutrient that leads to the production of amino acids and nitrogen compounds and as signal molecules in plant development and metabolism. Hence, PAW treatment for 20 min was selected for the irrigation water throughout the whole plantation period of water spinach.

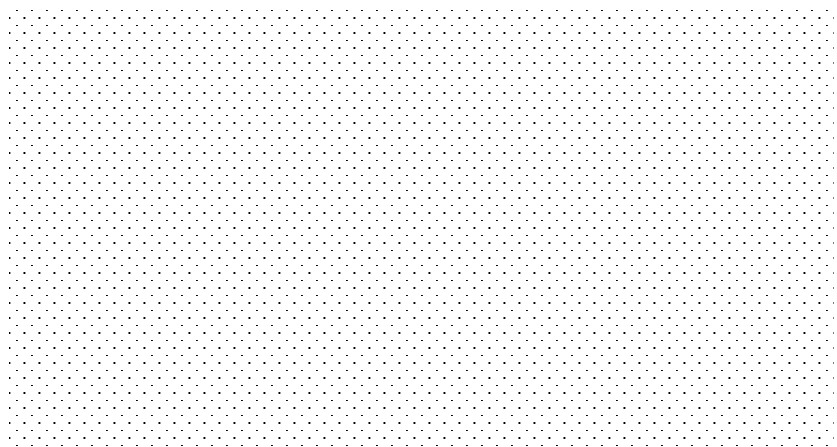

**Figure 2.** Concentration of nitrates ($NO_3^-$) produced in PAW under different treatment times ($n$ = 3).

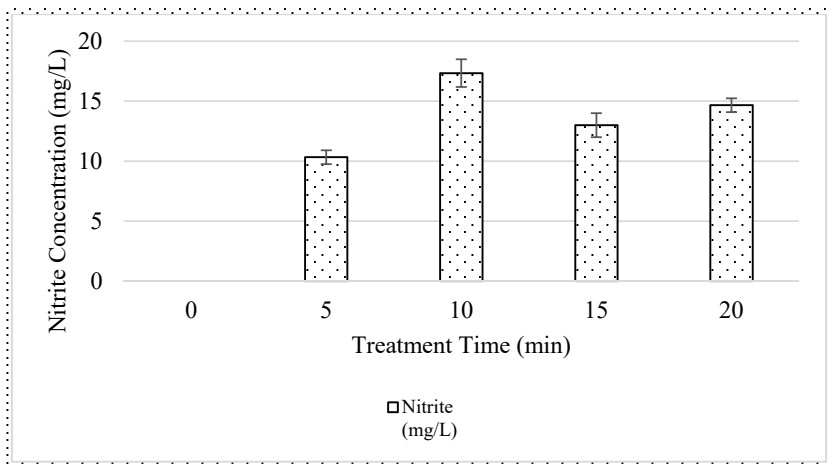

**Figure 3.** Concentration of nitrites ($NO_2^-$) produced in PAW under different treatment times ($n$ = 3).

### 3.2. Concentration of Heavy Metals in Man-Made Contamination Soil

The initial heavy metal concentration was determined before plantation. The Cd-added cultivation soil was expected to be 7 mg/kg W.W., while Pb-added cultivation soil was expected to be 700 mg/kg W.W. The expected heavy metal concentrations in the soil were selected to be 3-times higher than the reference of the monitored and controlled standard for metals in soil stated by the Taiwan EPA; therefore, the added Cd concentration was 18.18–22.28 mg/kg dry weight (D.W.), and the Pb concentration was 2064–2492 mg/kg D.W (Table 1).

**Table 1.** The metal concentrations of control soil, Cd-added soil and Pb-added soil (mg/kg D.W.).

| Treatment | n | Cd Concentration | | Pb Concentration | |
| --- | --- | --- | --- | --- | --- |
| | | **Control Soil** | **Cd-Added Soil** | **Control Soil** | **Pb-Added Soil** |
| NTS + NTW | 3 | ND-<LOQ | 18.18–20.43 | 2.60–3.17 | 2222–2492 |
| NTS + PAW | 3 | <LOQ | 19.06–19.53 | 2.36–3.77 | 2064–2470 |
| PTS + NTW | 3 | ND-<LOQ | 20.93–22.28 | 1.84–2.32 | 2182–2429 |
| PTS + PAW | 3 | ND-<LOQ | 21.17–21.92 | 1.77–2.80 | 2086–2448 |

Note: The results are represented as min–max value. ND: Not detected; LOQ: Limit of quantification.

### 3.3. Harvested Water Spinach

The total weight of harvested water spinach edible portions was higher in water spinach planted in soil with Pb contaminants, but the total quantity of harvested water spinach decreased with either plasma treatment of the seeds or when PAW was used as the irrigation water (Table 2). The sequence for the height of the water spinach planted in Cd-added soil was as follows (Figure S5): PTS + NTW > Control > NTS + PAW > PTS + PAW. As the average heights of water spinach in NTS + PAW and PTS + PAW treatment groups were lower than those in the control group in Cd-added soil, it can be concluded that PAW usage and combined treatments of PTS and PAW inhibited plant growth in terms of plant height in this study. The sequence for the height of water spinach planted in Pb-added soil was as follows (Figure S6): PTS + PAW > PTS + NTW > Control > NTS + PAW. Overall, PAW usage may inhibit growth in terms of plant height when Pb was present in the soil.

**Table 2.** The total weight of harvested edible parts of water spinach (in term of fresh weight).

| Treatment | n | Total Weight (g) of Water Spinach Grown in | | |
| --- | --- | --- | --- | --- |
| | | **Control Soil** | **Cd-Added Soil** | **Pb-Added Soil** |
| NTS + NTW | 3 | 9.24 | 9.43 | 7.28 |
| NTS + PAW | 3 | 7.72 | 7.40 | 6.38 |
| PTS + NTW | 3 | 9.29 | 6.48 | 7.38 |
| PTS + PAW | 3 | 5.91 | 7.43 | 8.39 |

Note: The data shown are the sum values of three repeats.

### 3.4. Concentration of Heavy Metals in Water Spinach

To investigate the effectiveness of plasma treatment for reducing the accumulation of heavy metals in vegetables, the heavy metal concentration in water spinach was analyzed. The Cd concentration of water spinach planted in the control and Cd-added soil group are presented in Table 3. When the water spinach was planted in soil with a Cd contaminant, the Cd accumulated by the water spinach was reduced by either plasma treatment on seeds or by using PAW as the irrigation water, but not by both. The Cd concentration of the water spinach in NTS + PAW, PTS + NTW, and PTS + PAW declined by 17.8%, 22.9%, and 4.5%, respectively, when compared with the group without any treatment, especially PTS + NTW group showed the best outcome on declining Cd accumulation in water spinach. Hence, the plasma treatment may demonstrate positive effects for reducing Cd accumulation in water spinach, which is consistent with the results of Kabir et al., who reported that Ar/O$_2$- and Ar/Air plasma-treated plants exhibited significantly reduced Cd concentrations in the shoot and root in wheat crop when reducing the expression of Cd transporters (*TaLCT1* and *TaHMA2*) in the root [24]. Meanwhile, Table 3 depicts the results of water spinach planted in soil with Pb contaminant. The Pb concentration of water spinach in NTS + PAW and PTS + NTW decreased by 0.57% and 1.42%, respectively. Moreover, PTS + PAW treatment on water spinach increased the Pb accumulation in the water spinach by 64.8%. Hence, individual plasma treatment of seeds or water did not demonstrate any obvious positive effects for reducing Pb accumulation in water spinach.

From a different perspective, although the concentration of Cd in the Cd-added soil (18.18–20.43 mg/kg D.W.) was approximately 100 times lower than the concentration of Pb in the Pb-added cultivation soil (2064–2492 mg/kg D.W.), the Cd concentration in the water spinach (12.1–15.7 mg/kg D.W.) planted in the Cd-added soil was greater than the Pb concentration in the water spinach (6.94–11.6 mg/kg D.W.) planted in the Pb-added soil. This result was consistent with that of Intawongse and Dean [25], who reported increasing uptake of Cd, Cu, Mn, and Zn by plants correlating with increasing concentration in soil, while the uptake of Pb was poor.

**Table 3.** The metal concentrations of water spinach planted in control, Cd-added soil and Pb-added soil (in terms of dry weight).

| Treatment | Cd Concentration | | Pb Concentration | |
| --- | --- | --- | --- | --- |
| | Control Soil (mg/kg D.W.) | Cd-Added Soil (mg/kg D.W.) | Control Soil (mg/kg D.W.) | Pb-Added Soil (mg/kg D.W.) |
| NTS + NTW | 0.265 | 15.7 | 0.703 | 7.04 |
| NTS + PAW | 0.182 | 12.9 | 0.246 | 7.00 |
| PTS + NTW | 0.215 | 12.1 | 0.284 | 6.94 |
| PTS + PAW | 0.207 | 15.0 | 0.697 | 11.6 |

Note: Samples used in the analysis were the mixtures of 3 pots of plants.

The combined effect of plasma-treated seeds and water may depend on the nature of the seeds [26]. Ahn et al. [27] conducted an experiment to compare various plasma conditions with different plasma generators on corn seeds grown in six separated locations in Illinois. The effect of plasma treatment on corn yield was affected by weather and soil quality. The $Ar/O_2$ plasma affected the wheat plant growth and development by increasing the ascorbate peroxidase (APX) activity in shoots and significantly increased in superoxide dismutase (SOD) activity and the TaSOD expression in the roots of wheat [28]. Safari et al. [29] reported that plasma treatment of 1 min resulted in an improved effect on the total leaf area, shoot, and root lengths, although plasma treatment for 2 min significantly impaired the growth, reducing the total biomass. The high intensity of plasma treatment could cause some changes in the gene expression, inhibiting the growth rates and inducing morphological and physiological differences [29]. According to some studies in the literature, and based on our results, the non-significant results with respect to plant growth following watering with PAW may be affected by metals in the soil, the cultivation environment, and plasma treatment of the seeds.

With respect to heavy metals, the presence of Pb in soil reduced the yield of water spinach, while Cd in soil did not affect the yield relative to that in the untreated group (NTS + NTW). The group with the PTS + PAW combination only showed a positive effect in terms of the total weight of the water spinach planted in Pb-soil.

In a nutshell, the average height and total weight of the water spinach were affected by the presence of heavy metals in soil. The total weight and the average plant height were restricted by the high concentration of Pb. On the other hand, these restricted parameters can be overcome or improved through plasma treatment. In this study, plasma treatment did not help to improve the product yield of water spinach planted in Cd-added soil, but it was able to increase the average plant height of water spinach after seed treatment. Finally, in the PTS + NTW group, the growth of the water spinach was increased in terms of both total weight and average height compared to that in the control soil.

Heavy metals enter the plants through soil uptake and accumulates in different organs, which may reduce plant growth and productivity, while some metabolic processes are also associated with Cd toxicity and its tolerance [30]. A high concentration of Pb can cause several toxic symptoms in plants and may lead to growth retardation in plants, the inhibition of photosynthesis, root blackening, changes in hormonal status, disturbance of mineral and water balance, and other symptoms [31]. Hence, the unsatisfactory plant growth observed in this study may be an effect of heavy metal toxicity.

As previously mentioned, the concentration of heavy metals in Cd-added and Pb-added cultivation soil was extremely high, which may have led to the restriction of plant growth. Since the heavy metal concentration used in this study was high and the overall results of plant growth were not consistent with those in some other works in the literature (Table 4), in the future, the plantation environment could be changed and fertilizer could be used to supply some nutrients that could not be provided by PAW in order to support the growth of water spinach.

**Table 4.** Summary of plasma treatment applications in seed germination and plant growth.

| **A. Plasma Treatment on Seeds** | | | | **Outcomes** | |
| **Author** | **Target** | **Plasma Generation** | **Parameters** | **Seed Germination** | **Plant Growth** |
|---|---|---|---|---|---|
| Ling et al. [11] | Oilseed rape (*Brassica napus* L. cv Zhongshuang 9) | Low-vacuum helium cold plasma (Radio frequency discharge) | Helium gas 100 W, 15 s, 13.56 MHz, 150 Pa | Positive | Positive |
| Jiang et al. [32] | Tomato (*Solanum lycopersicum* L. cv. Shanghai 906) | Inductive helium capacitive coupled plasma (CCP), Computer-controlled plasma treatment apparatus HD-2N (Radiofrequency) | Helium gas 80 W, 15 s, 13.56 MHz, 150 Pa, 3.5 eV (electron temperature) | Positive | Positive |
| Ling et al. [33] | Peanut (*Arachis hypogaea* L. cv. Eyou 7) | Inductive helium discharge with HD-2N units (Radiofrequency generator) | Helium gas, 0–120 W, 15 s, 13.56 MHz, 150 Pa | Positive | Positive |
| Li et al. [34] | Soybean (*Glycine max* L. Merr cv. Zhongdou 40) | Inductive helium discharge with computer-controlled plasma treatment apparatus HD-2N units (Radiofrequency generator) | Helium gas 0–120 W, 15 s, 13.56 MHz, 150 Pa | Positive | Positive |
| Zhang et al. [35] | Maize, peppers, wheat, soybeans, tomatoes, eggplants, pumpkins etc. | Electromagnetic shielding and suspension electrode technology; High (glow) radiofrequency discharger produces plasma | Air, 13.56 MHz, 80–180 W, 30–200 Pa, 5–90 s | Positive | Positive |
| Saberi et al. [36] | Wheat (*Triticum aestivum* L.) | Non-thermal Radio Frequency Plasma | Air, 80 W, 13.56 MHz, 0.1 mbar, 0–240 s | - | Positive |
| Jiang et al. [37] | Wheat (*Triticum* spp.) | Cold plasma generator | Helium gas, 60–100 W, 15 s, $3 \times 10^9$ MHz, 13 eV, 150 Pa | Positive | Positive |
| Safari et al. [29] | *Capsicum annum* PP805 Godiva | DBD plasma | Argon gas, 23 kHz, 11 kV, 80 W, 94.98 $cm^2$ plasma treatment areas, 0.84 W/cm power density, 0–2 min | - | Positive |

**Table 4.** *Cont.*

| A. Plasma Treatment on Seeds | | | | | |
|---|---|---|---|---|---|
| | | | | **Outcomes** | |
| **Author** | **Target** | **Plasma Generation** | **Parameters** | **Seed Germination** | **Plant Growth** |
| Mihai et al. [38] | Radish seed | Non-thermal plasma-surface discharge | Air, 15 kV, 2.7 W, 20 min, Gas flow rate = 1 L/min | No Change | No Change |
| de Groot et al. [39] | Cotton seeds variety Sicot 74BRF | Cold atmospheric-pressure plasma (CAP) | Air/Argon gas; Flow rate = 1 L/min, AC power supply, 1 kHz, sine wave with 38 kVpp for air and 11 kVpp for argon, treatment time: 0, 3, 27 min with dry air and 81 min with argon gas | Positive | Not significant but positive for Air-27 min and Ar-81 min, and negative for Air-3 min |
| Iranbakhsh et al. [40] | Wheat (*Triticum aestivum* L. cv. Parsi) | Dielectric barrier discharge (DBD) | Nitrogen and Helium gas, 20 kHz, 15 kV, 100 W, 254.3 cm$^2$; 0.4 W/cm; 100 Pa, 15, 30, 60, 120 s, repetition with 1, 2, 4 times with 24 h intervals | - | Positive |
| Pawlat et al. [41] | *Lavatera thuringiaca* L. | Gliding arc reactor | Nitrogen gas (8 L/min), 680 V, 50 Hz, 33 mA, 40 W, 1, 2, 5, 10, 15 min | Positive | - |
| Pawlat et al. [10] | *Lavatera thuringiaca* L. | Dielectric barrier discharge (DBD) plasma jet | Helium: 1.6 dm$^3$/min, Nitrogen: 0.03 dm$^3$/min, 3.7 kV; 17 kHz; mean of 6 W, 1, 2, 5, 10, 15 min | Positive | - |
| Rahman et al. [28] | Wheat (BARI Gom 22) | Low pressure dielectric barrier discharge (DBD) | Ar60%/Air40%; Ar60%/Oxygen40%, 5 kV, 4.5 kHz, ~45 W, 90 s | Positive | - |
| B. Plasma Treatment on Seeds through Aqueous Media | | | | | |
| | | | | **Outcomes** | |
| **Author** | **Target** | **Plasma Generation** | **Parameters** | **Seed Germination** | **Plant Growth** |
| Zhou et al. [21] | Mung bean seeds (*Vigna radiata* Linn. Wilczek) | Atmospheric pressure microplasma array | He, N$_2$, artificial Air, O$_2$, (2 standard liters per min), 36 microplasma jet units, 4.5 kV, 9.0 kHz, 25 W, 10 min | Positive | Positive |
| Liu et al. [12] | Tomato, Lettuce, Mung bean, Sticky bean, Radish, Dianthus, Mustard, Wheat | Dielectric barrier discharge (DBD) | N$_2$, O$_2$, Synthetic Air, 1.5 L/min, 0–18 kV, 500 Hz, Power consumption: ~2.5 W, Vpp = 20 kV, 2, 4, 6 min | Positive | Positive |

| C. Combination: Plasma Treatment on Seeds and Water | | | | | |
|---|---|---|---|---|---|
| | | | | **Outcomes** | |
| **Author** | **Target** | **Plasma Generation** | **Parameters** | **Seed Germination** | **Plant Growth** |
| Bafoil et al. [42] | *Arabidopsis thaliana* (Early stage) | Floating electrode dielectric-barrier discharge (FE-DBD) | Ambient air, 10 kV for the voltage, 9.7 kHz for the frequency and 1 us for the pulse duration, 15 min | Positive | - |
| | Distilled water and tap water (Later stage) | Plasma jet, the floating electrode-dielectric barrier discharges (FE-DBD) | Helium gas, 10 kV, 9.7 kHz, 1μs pulse duration, 3 L/min, 15 min, 30 mL water used in each treatment | - | Positive |
| Sivachandiran et al. [26] | Water | Cylindrical double DBD reactor in air under atmospheric pressure and room temp | Synthetic Air (Air Liquide), Flow rate = 1 L/min, Pulse width: 120 ns; 21 kV; 2.4 A; 400 Hz; Max energy: 7 mJ, 250 mL DI water activated for 15 min and 30 min | Positive when treat on dry seeds, no significant influences on wet seeds | Negative for stem length on P-10 min seeds and P-20 min seed + PAW-30 min |
| | Radish, Tomato, Sweet Pepper seeds | Plate-to-plate double DBD reactor in air under atmospheric pressure and room temp | Synthetic Air (Air Liquide), Flow rate = 1 L/min, Pulse width: 120 ns; 21 kV; 18 A; 200 Hz; Max energy: 57 mJ, 10 min; Plasma discharge volume: 130 W/cm | | |

*3.5. BCF of Vegetables Planted in Contaminated Soils*

Bioaccumulation or bioconcentration refers to an increase in particular chemicals or elements in organisms over a period of time, which may harm the organism itself or its predators [43]. The BCF of Cd and Pb were calculated, and the results are depicted in Table 5 and Figure S7. Overall, BCF of Cd in water spinach (0.543–0.864) was higher when compared to that of Pb (0.003–0.006). The treatment of NTS + NTW indicated the highest BCF of Cd in water spinach, followed by PTS + PAW and NTS + PAW, while PTS + NTW showed the lowest BCF of Cd in water spinach. The BCF of Cd in water spinach decreased from 0.864 to 0.543 following the plasma treatment, while the BCF of Pb was low and did not show any significant changes. These results are similar to those reported by Huang et al. [44], which showed that the BCF of Cd in two water spinach cultivars was much greater than that of BCFs of Pb, irrespective of whether it was in the leaf, stem, or root.

Saad et al. [45] showed that the bioaccumulation factor (BAF) of Cd in water spinach was slightly higher than that of Pb, albeit not significant.

**Table 5.** Bioconcentration factors (BCF) of Cd and Pb in water spinach.

| Treatment | *n* | Cd | Pb |
|:---:|:---:|:---:|:---:|
| NTS + NTW | 3 | $0.819 \pm 0.048$ | $0.003 \pm 0.000$ |
| NTS + PAW | 3 | $0.669 \pm 0.008$ | $0.003 \pm 0.000$ |
| PTS + NTW | 3 | $0.564 \pm 0.019$ | $0.003 \pm 0.000$ |
| PTS + PAW | 3 | $0.695 \pm 0.012$ | $0.005 \pm 0.000$ |
| *p*-value | | 0.016 * | 0.092 |

Note: * $p < 0.05$, Kruskal–Wallis Test. Samples used in the analysis were the mixtures of three pots of plants.

Sengar et al. [46] mentioned that several factors may affect the amount and speed of uptake and transpiration of Pb into plants, such as soil particle size, soil cation-exchange capacity, root surface area, organic matter content, root exudation, mycorrhizal transpiration rate, and soil pH values. In addition, Pb absorption by plants may also be affected by the life stage and passive environmental factors. At the lethal concentration of Pb, the barriers in the root endodermis will be broken and cause lead to enter the vascular tissues. This point may further affect the mineral nutrition and water balance, inactivate the enzyme activities, change the hormonal status, and affect the membrane structure and permeability [46]. Pourrut et al. [47] reported that once Pb enters the root system, approximately 95% or more of Pb is accumulated in the roots and only a small fraction is translocated to the aerial plant section. This phenomenon might be different for each kind of heavy metal, and is specific to Pb. Pb is blocked in the endodermis by the Casparian strip, which is followed by symplastic transport. In addition, the majority of the Pb entering the root is removed by the plant's detoxification systems. Some plant species are able to tolerate higher concentrations of Pb ions due to various detoxification mechanisms, including selective metal uptake, excretion, complexation by specific ligands, and compartmentalization.

According to a report by Wang et al. [48], lower pH influences the metal uptake of plants because Cd and Zn are more soluble at lower pH; this may be an effective strategy for enhancing the phyto-extraction of metals from soil. Ye et al. [49] also reported that soil pH and organic matter content are important in affecting the Cd availability in soil. Both of these elements determined the adsorption of Cd at soil binding sites, as well as its solubility and mobility in soil. The determination of soil pH was not performed in this study; hence, the mechanisms of low pH of PAW are expected to increase the metal solubility, while increased uptake by plants needs to be determined in the future.

In this study, the accumulation of Cd in water spinach was reduced, but the Pb concentration in water spinach in the PTS + PAW treated group was enhanced. Our results for Pb concentration increment in water spinach were similar to those of Kabir et al. [24]. Meanwhile, our study reported a reduction in Cd concentration in water spinach through the inhibition of expression of Cd transporter in the root by plasma treatment, but an increase in Pb concentration may be related to the activation of Pb transporter in the root of water spinach by plasma treatment on seeds and PAW irrigation.

## 4. Conclusions

The accumulation of heavy metals by water spinach was found to be dependent on the type of heavy metals and their concentration in soil. A higher concentration of Cd was accumulated in water spinach compared to that of Pb. The PTS + NTW treatment group demonstrated the highest reduction in Cd accumulation in water spinach. However, plasma treatment of seeds or PAW irrigation was able to reduce the accumulation of heavy metals in water spinach, except for water spinach in the group of PTS + PAW grown in Pb-added soils. The limitations of this study, such as the insufficiency of the planting space, the fixed condition of the PAW generator, and the small sample size, could have led to imperfections in the study. An increased sample size in order to achieve a more



powerful analysis, modification of the optimal parameters (plasma power, treatment time, etc.) of the PAW treatments, and the selection of a more suitable PAW generator will all be taken into account in future studies. Meanwhile, since the heavy metal concentration used in this study was high and the plant growth in the current study was not consistent with that described in some other studies in the literature, this study on the effects of various concentrations of Cd and Pb on the growth and development of water spinach is a prerequisite for research into the combined influence of PAW and heavy metals on this plant species.

**Supplementary Materials:** The following are available online at https://www.mdpi.com/article/10.3390/app11115304/s1. Table S1 The physiochemical properties of PAW under different treatment time, Figure S1 The pH values of PAW under different treatment time ($n = 3$), Figure S2 The ORP values of PAW under different treatment time ($n = 3$), Figure S3 The electrical conductivity of PAW under different treatment time ($n = 3$), Figure S4 Concentration of $H_2O_2$ produced in PAW at different treatment time ($n = 3$), Figure S5 The average height of water spinach planted in Cd-added soil under different treatments, Figure S6 The average height of water spinach planted in Pb-added soil under different treatments, Figure S7 Bioconcentration factors of Cd and Pb in water spinach.

**Author Contributions:** Conceptualization, H.-L.C.; methodology and processing, T.-K.K., C.-Y.H., C.-M.L. and H.-L.C.; chemical analysis, T.-K.K.; resources, C.-Y.H., C.-M.L. and H.-L.C.; writing-original draft preparation, T.-K.K. and C.-Y.H.; writing-review and editing, C.-M.L. and H.-L.C.; supervision, C.-Y.H.; project administration, C.-M.L. and H.-L.C.; funding acquisition, H.-L.C. All authors have read and agreed to the published version of the manuscript.

**Funding:** This work has been supported by Lab of Hsiu-Ling Chen.

**Institutional Review Board Statement:** Not applicable.

**Data Availability Statement:** This study did not report any data.

**Acknowledgments:** The authors acknowledge that the plasma jet system was designed and developed by the Mechanical and Mechatronics Systems Research Lab, Industrial Technology Research Institute (ITRI) (Hsinchu, Taiwan).

**Conflicts of Interest:** The authors declare no conflict of interest.

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
