# Peer review of "The Effects of Plasma-Activated Water on Heavy Metals Accumulation in Water Spinach"

_applsci, doi:10.3390/app11115304_

Round 1

Reviewer 1 Report

The manuscript is significantly improved in the revised version. All my comments to the previous version were taken into account and sufficiently incorporated into the text. Therefore, I suggest the manuscript for publication.

Author Response

Thanks for your essential recognition.

Reviewer 2 Report

Generally, authors tried to provide decent answers and revisions. However, several points still need to be addressed.

  1. In this study, it is not clear if authors focus on plasma effect on metal absorption ability or plant performance under metal stress. Plasma treatment condition used in the study (plasma power, treatment time, etc) does not seem to be optimal (inhibit in NTS+PAW, PTS+PAW) for spinach growth in normal environment (no heavy metal stress). This would affect metal absorption. It would be better to discuss the interrelation among plasma effect, growth, and metal absorption.
  2. In table 2 and 3, indicate standard deviation or error in each value.
  3. I still think that table 4 is not necessary. Studies on seed germination and plant growth are much more than those stated in table 4. It would better to mention in the text as references.
  4. English has to be re-checked, particularly in modified area.

Author Response

1. In this study, it is not clear if authors focus on plasma effect on metal absorption ability or plant performance under metal stress. Plasma treatment condition used in the study (plasma power, treatment time, etc) does not seem to be optimal (inhibit in NTS+PAW, PTS+PAW) for spinach growth in normal environment (no heavy metal stress). This would affect metal absorption. It would be better to discuss the interrelation among plasma effect, growth, and metal absorption.

Response: We have discussed how the metals influence the plant growing as “Heavy metal enters the plants through soil uptake and accumulates in different organs, which may reduce the plant growth and productivity, and some of the metabolic processes are also associated with Cd toxicity and its tolerance [31]. A high concentration of Pb can cause several toxic symptoms in plants and may lead to growth retardation of plants, the inhibition of photosynthesis, roots blackening, changes in the hormonal status, disturb minerals and water balance, and other symptoms [32]. Hence, the unsatisfied plant growth in this study may be affected by the heavy metal toxicity. As previously mentioned, the concentration of heavy metals in Cd-added and Pb-added cultivation soil was extremely high, which may have led to the restriction of plant growth. Since the heavy metal concentration used in this study were high and the overall results of plant growth was not consistent with those in some other work of literatures (Table 4), in the future, the plantation environment could be changed and fertilizer can be used to supply some nutrients that could not be provided by PAW to support the growth of water spinach.” in Line 377-390 of Page 10. In addition, we also discussed the reason of PAW may affect metals uptake by the plants as “According to a report by Wang et al. [49], lower pH influenced the metal uptake by plant since Cd and Zn are more soluble at decreasing pH; this may be an effective strategy to enhance phyto-extraction of metals from soil. Ye et al. [50] also reported that soil pH and organic matter content are important in affecting the Cd availability in soil. Both of the elements determined the adsorption of Cd at soil binding sites, solubility, and mobility in soil. The determination of soil pH was not performed in this study; hence, the mechanisms of low pH of PAW are expected to increase the metal solubility, while increased uptake by plants needs to be determined in the future.” in Line 425-432 of Page 12. We agreed with the opinion of the reviewer mentioned that Plasma treatment condition used in the study (plasma power, treatment time, etc) does not seem to be optimal for spinach growth in normal environment without heavy metal stress, therefore, the sentence “To increase the sample size to achieve the power analysis, to modify the optima parameter (plasma power, treatment time, etc) of the PAW treatments, and to choose a more suitable PAW generator will be taken into account in future studies.” was addressed in Line 450-452 of Page 13.

2. In table 2 and 3, indicate standard deviation or error in each value.

Response: In table 2 and 3, there is only a mixed sample and we didn’t show any standard error or standard deviation. However, in figure 2 and 3, we showed standard deviation on there.

3. I still think that table 4 is not necessary. Studies on seed germination and plant growth are much more than those stated in table 4. It would better to mention in the text as references.

Response: Sure, we believe that more researches may in relevant to seed germination and plant growth as the reviewer mentioned. However, in table 4, we try to organize kinds of research in initiating us to design the current study to use PAW for seed germination and for irrigating while soil with metal contaminants. Therefore, we strongly suggested keeping the table in this manuscript.

4. English has to be re-checked, particularly in modified area.

Response: Now, we have checked the grammar in whole manuscript by the software on line (https://app.grammarly.com/ddocs/1173010814). All the corrections have been marked in red in the text.

This manuscript is a resubmission of an earlier submission. The following is a list of the peer review reports and author responses from that submission.

Round 1

Reviewer 2 Report

Non-thermal plasma (NTP) is an alternative/complementary technology, for stimulating growth and reducing pathogen/chemical contamination in plant seeds. The NTP is also effective as pre-treatment method for seeds for stimulating germination and growth by means of  reducing the microbial layers on seeds and changing their water  absorption properties. Some evidence exists to support that the radical particles from a water-based plasma can have a direct impact on the stress controlling chemical mechanisms. The submitted manuscript concerns the potential application of non-thermal plasma in decreasing of heavy metal accumulation in vegetables cultivated in contaminated soil by initiating the growth of plants.

Specific comments:

1.Introduction

page 3

RON - this abbreviation should be explained

2. Material and Methods

2.1. Man-made contaminated soil (page 3)

Why the concentrations: 700 mg/kg wet weight (W.W.) of lead (II) acetate and 7 mg/kg W.W. of cadmium acetate were used to the experiments?

What criteria were considered for the selection of metal concentrations?

2.3. Plasma treatment on seeds and irrigation water (page 4)

For seeds’ treatment, 15 seeds were selected randomly and placed in a beaker containing 200 mL of RO water. One electrode was immersed under water and the power was set to 60 W for plasma generation. The seeds were treated for 7 min….

For the control group, the seeds will be placed in the water for 7 min without any plasma application.”

Why seeds were not placed in freshly prepared PAW for 7 min (or other time), but submerged in RO water and treated with The current (60 W).

The control group is not properly selected. Is it possible to proof  that the current (60W) have no direct effect on the seeds via electrical stimulation?

RO - this abbreviation should be explained.  

2.4. Plantation of water spinach (page 4)

“For the seedling preparation, the treated and untreated seeds were then placed on a wet filter paper in a Petri dish for a germination period of 5 days. Six of the best-grown seedlings were selected and moved to each pot for growing.”

According to the references, quoted in the manuscript,  plasma treatment is able to improve the germination. Why the influence of plasma on the germination of water spinach was not taken into account in the research and the data was not presented in the results.

2.5.3. Preparation of vegetable samples (page 5)

The samples collected from the same treatment combination were maintained in the same zipper bag to make it as a mixed sample from 3 pots of a plant.

Does it mean that determination of heavy metals was made only once for each experimental group?

3. Results and Discussion

3.1. Physiochemical properties of irrigation water (page 6)

Error! Reference source not found.

What does it stand for?

“NO3 plays important roles as nutrient that leads to the production of amino acids and nitrogen com-

pounds and as signal molecules in plant development and metabolism. Hence, PAW treatment for 20 min was selected as the irrigation water in the whole plantation period of water spinach.”

Why PAW, prepared in the same way, was not chosen for seeds treatment?

Figure S1

Can it be concluded from the Figure that the PAW obtained after treating RO with current (60 W) for 7 min (seed treatment) has physicochemical properties similar to untreated RO, and to a much lesser extent to PAW obtained after treatment of RO with current (60 W) for 20 min?

How long does PAW maintain its physicochemical properties?

How would RO water, adjusted to pH 3.0 (with acid), affect the growth of water spinach? Were such control experiments carried out?

3.2. Concentration of heavy metals in man-made contamination soil (page 7)

“The expected heavy metals concentration in soil was selected in 3-times greater concentration than the reference of monitored and control standard of metals in soil by Taiwan EPA;”

Why the expected heavy metals concentration in soil was selected as 3-times greater  (not 2-times) than the control standard of metals in soil by Taiwan EPA?

3.3. Water spinach harvested (page 8)

Table 3.

“The data shown are the sum values of 3 repeats”

Does it mean that the total weight of harvested water spinach edible parts was measured only once and statistical analysis of tested groups cannot be made?

“The total weight of harvested water spinach edible portions was higher in water spinach planted in soil with Pb contaminants, but the total quantity of harvested water spinach decreased either plasma treatment on seeds or used PAW as the irrigation water (Table 3).

The total weight of harvested water spinach edible portions was lower in water spinach planted in soil with Pb contaminants (control soil 9,24 g; Pb-added soil 7.28 g) but the total quantity of harvested water spinach decreased either plasma treatment on seeds or used PAW as the irrigation water (Table 3).”

Which differences between the mean height of the water spinach planted in soil with Cd (Figure S4) and the soil with lead (Figure S5) are statistically significant? What is the significant differences between  the bioconcentration factors of Cd and Pb (Figure S5)?

Why the last two figures from the supplementary data have the same numbering (5)?

Are the conclusions included in paragraph 3.2 reliable in the absence of statistical analyzes?

3.4. Concentration of heavy metals in man-made contamination soil (page 8-10)

Why two paragraphs 3.2. (page 7) and 3.4. (page 8) have the same title "Heavy Metal Concentration in Human Contaminated Soil"  The content of the paragraph 3.4. refers to the concentration of metals in water spinach.

Table 3. 4 and 5

“The data shown are the sum values of 3 repeats”

Does it mean that:

  • the metal concentrations in edible parts (Table 3),
  • the metal concentrations in edible parts (in terms of dry weight) (Table 4),
  • the bioconcentration factors (BCF) of Cd and Pb (Table 5)

were measured only once and statistical analysis of tested groups cannot be made?

Are the conclusions included in paragraph 3.4 reliable in the absence of statistical analyzes?

 “…Cd concentration of water spinach (12.1–15.7 mg/kg D.W.) planted in Cd-added soil was greater than Pb concentration in water spinach (6.94–11.6 mg/kg D.W.).”

“This result was consistent with that of Intawongse  and Dean [25], who reported the uptake of Cd, Cu, Mn, and Zn by plants in accordance with increasing concentration in soil, while the uptake of Pb was poor.”

In general relatively low concentration of Pb in the aboveground parts of plants may also be explained by the fact that  80–95% of lead taken up from soil solution can be stored in roots. In contrary Cd is a mobile element and plants can take up, transport and distribute it to all of their organs effectively.

“The determination of soil pH was not performed in this study; hence, the mechanisms of low pH of PAW is expected to increase the metal solubility, while increase uptake by plants need to be determined in the future.”

The determination of soil pH is crucial for the experiments presented in the manuscript as low pH may affect plant growth by itself and interfere with effects caused by PAW. Spinach was watered with PAW pH 3 for 5 weeks, in spite the fact  the optimal pH for this species is 6.5–7.  

“.. the concentration of heavy metals in Cd-added and Pb- added cultivation soil were extremely high, which may have led to the restriction of plant growth. Since the heavy metal concentration used in this study was high and the overall results of plant growth was not consistent with those in some other literatures (Table 6)…”

The preliminary study on the effects of various concentrations of Cd and Pb on the growth and development of water spinach is a prerequisite for research on the combined  influence of PAW and heavy metals on this plant species.

Reviewer 3 Report

Dear authors

Thanks for your paper. Unfortunately, however idea is interesting and important, there are too many points that must be improved, corrected, re-measured, etc. Some comments are given below, but this is not a full list what must be considered and improved. Based on this I’m recommending refuse submission and completely rewrite it and add other experiments before re-submission.

  1. The statement “ equal number of atoms and/or molecules in a charged or ionized material state“ is not true. This is valid about total charges including electrons and it is called quasineutrality.
  2. Common term is plasma activated water, not „plasma activation water“
  3. Statement „ The electrons receiving the high energy cause an increase in collisions in the gas.“ is not true.
  4. Examples given in Table 1 far of complete overview. There are many test of different studies about plasma or plasma activate water applications on plants and hundreds papers about seeds. It will be better to focus only on relevant papers with respect to the research presented in paper
  5. What kind of soil was used? There are many different with very different properties, so detailed soil description must be given to obtain relevant results.
  6. What RO water does mean?
  7. What electrode material and electrode configuration was used? I’m not sure about any discharge presence under given conditions. Additionally, wat was water conductivity?
  8. Number of used seeds and later plants is absolutely non-sufficient for any bio experiment.
  9. What were growing conditions?
  10. PAW contains also hydrogen peroxide that creates significant stress to seeds and plants. Did you determine it? What techniques were used for nitrides and nitrates determination? There are very strong interferences using some methods – see papers of P. Lukes.
  11. Table 6 should be moved into introduction. The detailed discussion of results with related literature must be done. And the paper related references should be kept.
  12. References 6 and 41 are the same. 16 and 19, too. Check all references for duplicities.

Reviewer 4 Report

The submitted manuscript presents results from experiments investigating the effects of plasma treated water (PAW) on the uptake of cadmium and lead in water spinach (Ipomoea aquatic Forssk.) growing on artificially Cd- and Pb-containing soils.

While PAW treatment of agricultural relevant plant species such as water spinach and its impact on heavy metal accumulation is of significant relevance for application-oriented research, plasma science and food industry, the presented study has strong substantive and linguistic weaknesses. Data and experiments are not properly presented and described. The content of many sections within the manuscript is confusing. The authors focus on facts and knowledge that are of no relevance to the study presented (e.g. table 1 and 6). Information, relevant for the presented study, is either not cited at all or is only mentioned superficially. Similar, important results are presented in the supplementary materials; instead, results less relevant to the study (fig. 2 and 3) are overrepresented in the manuscript.

First, the abstract is very vague and important findings are not mentioned. It should be clear from the beginning of the manuscript that several kind of PTW treatments and combinations have been performed, such as water spinach seeds treated with and sown in PTW and/or watering of growing plants with PTW during further growth in soil.

Second, since the manuscript is submitted to the section Food Science and Technology of applied sciences, a detailed description of the plasma source used to treat water for PTW generation is mandatory! It is curious to read how the authors describe published plasma sources and parameters from literature in detail without giving any useful information about the plasma source and parameters applied in their own study!

Third, many parts of the introduction are superfluous and not expedient in terms of the presented results. Presenting knowledge and research findings from other kind of plasma treatments mislead a future readership not familiar with cold plasma research. Thus, as the focus of the physical treatment method is plasma treated water (PTW, PAW), the introduction should focus mainly on this topic. Table 1 should be removed. Much more attention on findings and knowledge on PTW/PAW research should be made and described in detail. The introduction should address and focus on how PTW/PAW can be generated by different plasma sources e.g. which kind of plasma sources exist for treatment of different volumes in different speed? Which kind of water/solutions/liquids has been treated so far? What are the major physico-chemical characteristics of different kinds of PTW? What are the biological targets and treatment modes of PTW treatment? Which physiological effects have been observed so far? Examples and citations of PTW experiments presented in table 6 could be implemented to the introduction section.

Please note, nitrate and nitrite ions are not referred as RNS!

In general, the introduction section should contain a short informative overview on heavy metals in soils, and on how plants respond to heavy metals (e.g. accumulation, avoidance, adaptation strategies). What are the toxicity levels and allowed concentrations of heavy metal in vegetable foods?

There should also be more information given about the agricultural importance of water spinach, e.g. cultivation practice, harvesting time, market values (not only in Taiwan). How the plant is consumed and processed, e.g. raw like a salad, cooked or fermented?

Fourth, the material and method section is incomplete and confusing. Detailed description of plasma source (experimental setup, geometry, power supply, generator, etc.) and parameter settings is missing (e.g. voltage, AC vs DC, frequency, power density etc.)!

In section 2.3., what is the meaning of the abbreviation “RO water”?

A detailed description of how seeds are treated with PTW is missing; the scheme does not help for understanding at all. What was the temperature of water during plasma exposure? Furthermore, what is the meaning of “The seeds were treated for 7 min after a preliminary study revealed that it was the best parameter for high efficiency in germinating”? This data should be presented! What was the pH, the nitrate, nitrate and hydrogen peroxide concentration after seven minutes of treatment?

Method description for the analysis of pH, nitrate, nitrate and hydrogen peroxide concentrations is missing!

Which kind of soil was used e.g. containing peat, clay, humus etc. What was the origin of soil e.g. if commercial company name is missing.

Within section 2.5, it is not clear which parts of the plants were harvested and used for the analysis, e.g. roots and shoots. For analysis of the soil, how the roots of the plants were separated from the soil? These questions affects the results presented in tables 2-5 of section 3.

Reference(s) for the method applied to extract, prepare and determine heavy metals in soils is missing and description of method is incomplete. Information on parameter settings and emission lines are missing to describe ICP-OES method properly. Similar, information on technical parameter settings is missing for ICP-MS method.

Fifth, within the result and discussion section, numerous sentences are incomplete or have substantive semantic failures and citations have format errors. Results of published studies are cited, but they are not brought into connection with gained results.

Figures and some tables are not formally correct and do not correspond to good scientific practice. Information on concentration units, precise information on the number of samples and / or standard deviations including statistical evaluation are missing in several tables. In addition, there is no precise information at which time point samples (soil and plants) were harvested for analysis. Which parts of the plants were analyzed, roots, shoots or entire plants? In table 2, there is great doubt about the information on the statistics, the minimum and maximum values. What is the scientific meaning and the logical consequence of information “(ND - <LOQ)”? How a standard deviation value can be calculated if two of the three analyzed soil samples cannot be measured? How a statistical test can be performed with such low test numbers? Standard deviation and statistics are missing in tables 3 and 4. What is the meaning of „the sum value of 3 repeats” in table 3? Were three plants analyzed only?

The discussion is in parts contradictory and by not adequately scientifically sound!

The main result of the study, that PAW reduces the accumulation of Cd in spinach, is not trustable for following reason: The authors did not clearly address the effect of PAW on the soil itself. The simplest explanation of the observed results might be that PAW leads to a more efficient wash out of heavy metals from the soil. Thus, it is reasonable to ask how homogenous the heavy metals were distributed in the soil during watering procedure for five weeks. The authors stated in section 2.5.1 that content of heavy metals in soil were measured at the beginning and at the end of the experiment. Those data are from high importance and should be presented as it will show how much of the metals are washed out during plant cultivation because of watering procedure. Moreover, those data should be presented from both, the “RO” watered and the “PAW” watered heavy metal containing pots. How the authors can exclude that PAW contributes to a more efficient flush out of the soil!

Moreover, according to several published studies (e.g. reference 25; Ali et al., 2020) the pH of soil seems to have a significant impact on the abundance of heavy metals in plants, either by altered metal solubility or by physiological responses of plants to external pH. Thus, the authors should measure the pH of the soils at the beginning of the experiment and at the end of the experiment (after five weeks of planting and treatments).

For further reading, see: Ali, U., Zhong, M., Shar, T. et al. The Influence of pH on Cadmium Accumulation in Seedlings of Rice (Oryza sativa L.). J Plant Growth Regul 39, 930–940 (2020). https://doi.org/10.1007/s00344-019-10034-x

Furthermore, plant biomass parameters such as fresh and dry weight of shoot (and root), shoot (and root) length and leaf number needs to be added and/or presented more clear including statistical validation! A more precise evaluation of plasma effects and discussion in respect of heavy metal stress should be drawn by improved data presentation and additional data.

Finally, if result and discussion sections are combined, the text should be structured with subheadings, otherwise the text will be confusing and difficult to read.

All sections have to be fundamentally revised, a lot of information is missing and the data analysis and discussion is insufficient. The language needs to be fundamentally revised as well. A confusing mix of past, present and future tenses throughout the manuscript should be avoided.

First and foremost, the authors have to prove that accumulation of heavy metals in water spinach is significantly altered by PAW treatment. Data needs to be added to proof the scientific significance of the study upon effects on heavy metal accumulation affected by plasma.

Reviewer 5 Report

In this manuscript, authors described the effects of plasma activated water on heavy metal absorption and growth of water spinach. Here are comments.

  1. In Materials and Methods, information about plasma device is not enough. Is this plasma device already published? Then, add reference. Otherwise, it would be better to add physical characteristics of plasma source in text or supplementary information.
  2. In page 4, “The seeds were treated for 7 min after a preliminary study re-vealed that it was the best parameter for high efficiency in germinating.” Please add reference for this.  
  3. In Results and Discussion, what is “Error! Reference source not found”? Authors should make a correction for this.
  4. In table 3, does total weight mean fresh weight or dry weight? It would be better to indicate in table 3.
  5. In table 3, NTS+PAW and PTS+PAW seem to have negative effect on plant growth in control soil. pH of PAW (20 min treatment) was very low (table S1), and this PAW was applied three times per week. I think this must be no good on plants. Authors should discuss about this.
  6. I am not sure that Table 1 and 6 are necessary since this manuscript is research paper. It would be better to mention in the text of introduction and discussion.
  7. Table numbers are not in order.
